# Workload Is Associated with Anxiety and Insomnia Symptoms in an Italian Nationally Representative Sample of Public Health Medical Residents: The PHRASI Cross-Sectional Study

**DOI:** 10.3390/healthcare12222299

**Published:** 2024-11-17

**Authors:** Alessandro Catalini, Lorenzo Stacchini, Giuseppa Minutolo, Angela Ancona, Marta Caminiti, Claudia Cosma, Veronica Gallinoro, Valentina De Nicolò, Fabrizio Cedrone, Pamela Barbadoro, Vincenza Gianfredi

**Affiliations:** 1Department of Biomedical Sciences and Public Health, Section of Hygiene, Preventive Medicine and Public Health, Polytechnic University of the Marche Region, Via Tronto 10/a, 60126 Ancona, Italy; 2Department of Health Sciences, University of Florence, Viale Pieraccini 6, 50134 Florence, Italy; 3Department of Health Promotion, Mother and Child Care, Internal Medicine and Medical Specialties, University of Palermo, Piazza delle Cliniche 2, 90127 Palermo, Italy; 4School of Hygiene and Preventive Medicine, Vita-Salute San Raffaele University, 20132 Milan, Italy; 5Department of Medicine and Surgery—Sector of Public Health, University of Perugia, Piazzale Gambuli 1, 06100 Perugia, Italy; 6Department of Public Health and Infectious Disease, Sapienza University of Rome, Piazzale Aldo Moro 5, 00185 Rome, Italy; 7Hospital Management, Local Health Authority of Pescara, Via Fonte Romana, 8, 65122 Pescara, Italy; 8Department of Biomedical Sciences for Health, University of Milan, Via Pascal 36, 20133 Milan, Italy

**Keywords:** medical residents, healthcare workers, workload, anxiety, insomnia, academic settings

## Abstract

Background/Objectives: Mental health disorders pose a substantial challenge for healthcare workers, particularly in the post-COVID-19 era. Public health medical residents (PHRs) played a pivotal role during the pandemic and were significantly affected by the heavy workload. This study aims to uncover potential associations between workload characteristics and symptoms of anxiety and insomnia in Italian PHRs based on data collected in 2022 through the Public Health Residents’ Anonymous Survey in Italy (PHRASI) study. Methods: A total of 379 residents completed the self-administered questionnaire comprising the Generalized Anxiety Disorder-2 (GAD-2), and the Insomnia Severity Index (ISI). Results: While 36% PHRs reported anxiety symptoms (GAD-2 ≥ 3), 12% reported moderate-to-severe insomnia symptoms (ISI ≥ 15). The multivariate logistic regressions showed that a high work–life interference was associated with the presence of anxiety and insomnia symptoms, while attending two or more simultaneous traineeships was associated with insomnia symptoms. A high workload perceived was positively associated with both the mental health outcomes considered, while the perception of work environment manageability was negatively associated with them. Conclusions: These findings underscore the significant role of the workload in influencing the mental health status of PHRs and emphasize the importance of fostering a supportive work environment that prioritizes mental well-being

## 1. Introduction

Mental health disorders pose a major public health challenge, affecting more than 900 million people worldwide [1]. The spectrum of pathological conditions influencing mental health is wide, with depression and anxiety being the most widespread and impactful on the overall burden of disease [2]. At the same time, insomnia disorders are estimated to affect approximately 10% of the adult population, while an additional 20% experience occasional insomnia symptoms [3]. Typically, anxiety disorders involve persistent intrusive thoughts or worries, leading individuals to avoid specific situations, and may manifest also with physical symptoms like sweating, trembling, dizziness, or a rapid heartbeat [4]. Insomnia is characterized by difficulties in initiating sleep, maintaining it, or achieving restful sleep; it can interfere with daily activities and induce daytime sleepiness [5].

The prevalence of mental disorder symptoms is notably higher among healthcare professionals compared to the general population, and the COVID-19 pandemic has furthered this trend, leading to a significant increase in mental disorders among healthcare professionals both during and after the pandemic [6,7,8]. Among mental health conditions, many studies focused on anxiety and insomnia in healthcare workers. An umbrella review of 72 meta-analyses published in 2024 identified a pooled prevalence of anxiety and sleep disturbance, respectively, of 31.8% and 36.8% in this population [9]. Considering only physicians, these prevalence values were lower: 26.3% for anxiety and 30.6% for sleep disturbance.

Anxiety and insomnia are multifactorial conditions influenced by multiple interacting determinants that lead to symptom manifestation. Older adults and women are more affected. In particular, gender differences are notable, with women showing a significantly higher likelihood of developing insomnia (58% increased probability) and anxiety (70% increased probability) compared with men [10,11]. These differences have roots in psychosocial and biological factors [12,13]. Among the first ones, gender norms play an important role. Masculine traits such as instrumentality are positively related to better mental health outcomes, while feminine traits, such as kindness and sweetness, are associated with anxiety when a low self-esteem is also present [11,14]. Another psychosocial factor mentioned in the literature is the greater sense of family responsibility often felt by women, which can generate work–family conflict and may contribute to the higher prevalence of mental health conditions among women [15]. Among biological factors, genetic, hormonal and neuroanatomical characteristics were pointed out to explain the increased susceptibility to anxiety and insomnia symptoms in women [16,17]. Other determinants involved in the onset of anxiety and insomnia symptoms are stress and other mental health conditions, but also pain and chronic diseases like diabetes, cardiovascular illnesses, and asthma [18,19]. Unhealthy habits like smoking, sedentary behavior, alcohol misuse, and caffeine consumption also increase the risk of developing insomnia and anxiety symptoms [20]. A systematic review and meta-analysis in 2023 identified 17 influencing factors for sleep disturbance in healthcare workers during the COVID-19 pandemic [13]. Among these are frontline work and days served in frontline work, department of service, night shift, years of work experience, received psychological assistance, worry about being infected, and degree of fear with COVID-19. Some reviews underlined the extreme workload related to the pandemic response activities as one of the main determinants of sleep disorders in healthcare workers [21,22]. Likewise, anxiety symptoms were shown to be alarmingly high in this population during the pandemic [23]. Furthermore, physicians working long shifts showed significantly higher percentages of anxiety symptoms [24]. Although a widely accepted definition does not exist, workload can be defined as the amount of work a person has to do [25]. However, the literature tends to distinguish between the actual amount of work—measured, for example, in working hours or, for healthcare personnel, in the numbers of patients to care for—and the workload perceived by the individual, often referred to as mental workload [26,27]. Mental workload can be defined as “the perceived relationship between the amount of mental processing capability or resources and the amount [of objective work] required by the task” [28].

If some studies explore workload in healthcare workers or, more specifically, in physicians, the literature does not focus on how workload characteristics are distributed within the category of resident physicians. Evidence is also lacking on how workload affects the mental health of medical residents. In Italy, medical residents occupy a unique position, bridging both their educational training and the professional work environment. While enrolled in academic specialization programs organized by Italian postgraduate medical schools, they spend the majority of their time in traineeships mostly within National Health System services, assuming varying levels of responsibility and autonomy. Not only are they a significant group in terms of numbers, but, in recent years, they have also become a fundamental resource to support a National Health System burdened by a chronic shortage of staff. During the pandemic, Italian medical residents were largely involved in the implementation of control and response measures to COVID-19 [29]. In particular, public health medical residents (PHRs) are resident physicians who undergo a specialization course of four years, encompassing training in healthcare organizations, preventive medicine, and infection prevention and control techniques. During the pandemic, they made an essential contribution by participating in risk communication, contact tracing, and vaccination. Considering the impact of the COVID-19 pandemic on healthcare professionals’ mental health, the “Public Mental Health” working group of the medical residents’ Assembly of the Italian Society of Hygiene, Preventive Medicine, and Public Health designed the Public Health Residents’ Anonymous Survey in Italy (PHRASI) study, a nationwide cross-sectional study focused on the mental health status of Italian PHRs [30].

The purpose of the current study is to identify possible associations of the workload objective and perceived characteristics with symptoms of anxiety and insomnia in Italian PHRs during the COVID-19 pandemic. The analyses will account for possible confounders. Furthermore, a moderation analysis by sex will be conducted to assess the role of sex differences in the identified associations. This analysis will enable us to account for sex differences that were significantly seen to influence both the ways and the extent to which various determinants contribute to the development of mental health symptoms [31,32]. In order to contextualize the results, the study will also describe the workload characteristics and estimate the prevalence of anxiety and insomnia symptoms in PHRs.

Our hypotheses are as follows:Both objective and perceived unfavorable workload characteristics are associated with worse anxiety and insomnia outcomes in PHRs as already shown in some studies concerning healthcare workers [21,22,24];Sex has a role in moderating the association between the workload characteristics and the mental health outcomes mentioned above. This hypothesis is based on the well-documented influence of sex on the development of insomnia and anxiety symptoms [10,12];The prevalence of anxiety and insomnia symptoms in PHRs is equal or higher when compared to the one reported in healthcare workers. This hypothesis arises from the inherently stressful nature of medical residency, particularly during the COVID-19 pandemic, which, as noted above, required PHRs to actively engage in infection prevention and control efforts—like the whole healthcare workforce did—while simultaneously meeting academic and educational responsibilities. Furthermore, although geographically limited, some studies in Italy already showed a similar prevalence of symptoms between medical residents and healthcare workers in general [7,33].

Hypotheses 1 and 2 are synthetized in Figure 1.

The results of these analyses will help identify workload characteristics that can be addressed by specialization program directors and policymakers to create and sustain a safe and healthy work environment, thereby improving PHRs’ mental health.

## 2. Materials and Methods

The data for this analysis are derived from the PHRASI study, a nationwide cross-sectional survey conducted in Italian PHRs to explore their mental health status and its determinants. Methodological details have been previously described [30]. Briefly, data were collected between 14 June and 26 July 2022, through an 88-item anonymous questionnaire administered via Google Form (©2022 Google, Mountain View, CA, USA). The study targeted PHRs enrolled in the specialization course of Italian postgraduate Public Health schools. Participants were recruited through the Medical Residents’ Assembly of the Italian Society of Hygiene and Preventive Medicine using various communication channels. All questions in the questionnaire were mandatory to prevent missing data, and participation was voluntary without incentives. The variables extracted from the PHRASI dataset to meet the objectives of this study can be summarized in three groups: workload characteristics, mental health symptoms, and socioeconomic and lifestyle characteristics (covariates).

### 2.1. Workload Characteristics

The variables describing workload characteristics were selected from the PHRASI datasets and classified in two sub-groups: objective workload and perceived workload. The objective workload sub-group of variables contained information on the engagement of PHRs in an additional employment different from the postgraduate course (yes/no), the validity of the additional employment’s working hours for the achievements of the postgraduate course’s learning objectives (yes/no), total weekly working hours, the simultaneous attendance of two or more traineeships in different units (yes/no), and work–life interference. The latter was measured using a question derived from the Work-Related Stress Questionnaire (WRSQ), a 13-item questionnaire presented in a pilot study and recently validated through data obtained from the PHRASI study [34,35]. The Cronbach’s alpha for the WRSQ calculated on the PHRASI sample is 0.80 (95% CI = 0.77–0.83). Its questions investigate four dimensions of the work-sphere: the workplace characteristics, the job demand, the job support, and unpleasant workplace conditions. Question number 10 focuses on work–life interference, falls into the job demand dimension, and asks how often the work interferes with family, social, and personal needs. The five possible answers (never; rarely; sometimes; often; and always) have been grouped into “infrequent” (never to sometimes) and “frequent” categories (often to always).

Perceived workload sub-group of variables contained information on whether the PHRs were perceiving a high workload (yes/no) and on their Work Sense of Coherence (Work-SoC) manageability. The high workload perceived variable was derived from question number 4 of the WRSQ that asks whether the workload is excessive [34]. The five possible answers (never; rarely; sometimes; often; always) have been grouped in “no” (never to sometimes) and “yes” (often to always), whereas the Work-SoC manageability, defined as the extent to which a worker perceives that adequate resources are available to cope with the demands in the workplace, was derived from the subscale manageability of the Italian version of the Work-SoC questionnaire (Cronbach’s alpha calculated on the PHRASI sample = 0.90 (95% CI = 0.89–0.92) [36]. The manageability subscale score results from two questions that ask the participant to assess on a 7-grade Likert scale how much the current job and work situation in general is influenceable (question 4) and controllable (question 7).

### 2.2. Mental Health Symptoms

Mental health has been investigated by measuring symptoms of anxiety and insomnia. The Generalized Anxiety Disorder-2 (GAD-2) Italian translation was used to assess anxiety symptoms [37]. GAD-2 is the shorter version of GAD-7 and keeps only its first two questions that represent the core anxiety symptoms, regardless of the underlying specific diagnosis [38,39]. The questions ask how often the respondent experienced anxious symptoms during the previous two weeks. In clinical practice, a score of 3 or more highlights the need for further investigating the symptoms [40]. The Cronbach’s alpha of the GAD-2 calculated on the PHRASI sample is 0.85 (95% CI = 0.82–0.88). The Italian version of the Insomnia Severity Index (ISI) was used to determine the severity of insomnia symptoms [41,42]. It is a short, validated test created to assess the severity of insomnia’s daytime and nighttime components. For ISI, the Cronbach’s alpha calculated on the PHRASI study sample is 0.87 (95% CI = 0.85–089). There are seven questions and a final score. Insomnia is classified as not clinically significant from 0 to 7; subthreshold insomnia from 8 to 14; moderate-intensity clinical insomnia from 15 to 21; and severe clinical insomnia from 22 to 28.

### 2.3. Socioeconomic and Lifestyle Characteristics

Age in years, sex, economic status, and lifestyle characteristics from the PHRASI study dataset were used as covariates to correct the models. The economic status was assessed asking PHRs to express their capacity to make ends meet with their own income. Possible answers were grouped in “hardly” or “easily”. Lifestyle group of variables included two scores: one for alcohol abuse and the other for physical activity. Alcohol abuse was evaluated administering the Italian version of Alcohol Use Disorders Identification Test—Consumption (AUDIT-C) [43]. AUDIT-C is a three-item validated short version of the AUDIT questionnaire. Each item is attributed 0 to 4 points. A final score equal to or greater than five for males and equal to or greater than four for females indicates a possible risky consumption of alcohol. The dichotomous variable of risky alcohol consumption (yes/no) was used to adjust the models. The Italian and shortened version of the International Physical Activity Questionnaire (IPAQ) was used to measure physical activity [44,45]. The IPAQ is a validated seven-item questionnaire designed for use with adults (ages 15 to 69) that asks about the type (walking, moderate, and vigorous) and quantity (days per week and time per day) of physical activity in the previous seven days. The questions are designed to produce distinct scores for walking, moderate-intensity activity, and vigorous-intensity activity, in addition to a combined total score, to characterize the overall level of activity. In order to create a dichotomous variable to be used to adjust the models, according to the total score, the participants have been grouped into “inactive” (<700) and “active” (≥700) categories.

### 2.4. Statistical Analysis

Continuous variables were reported as mean and standard deviation (SD) for normal distribution or median and interquartile range (IQR) for non-normal distribution. Dichotomic and categorical variables were presented as both absolute frequencies and percentages. As appropriate, Chi-square or Fisher’s exact test were performed to evaluate the association between various groups of mental health symptom status and the workload characteristics of the sample. Student’s t-test or Wilcoxon–Mann–Whitney test, as appropriate, was performed to evaluate the difference in the distribution of continuous variables between the two groups. To assess the difference between groups with more than two groups, ANOVA or Kruskal–Wallis test, as appropriate, was performed. For the main analysis, the scores of the two mental health symptoms were grouped: for GAD-2, according to a cut-off of 3 to detect anxiety symptoms, and, for ISI, according to a cut-off of 15 to detect moderate to severe insomnia symptoms. Before performing the regressions, multicollinearity among independent variables was assessed calculating Pearson’s correlation coefficient [46]. A cut-off of 0.8 for the Pearson’s coefficient is generally used to assess collinearity and was adopted for our study [47]. Multiple logistic regression analyses were performed following a forward stepwise approach. First adjustment only for sex and age was applied (model 1). Subsequently, further adjustments for sex, age and economic status (model 2) and for sex, age, physical activity, and alcohol consumption (model 3) were included in the analysis. Adjusted odds ratios (aORs) and their 95% confidence interval (95% CI) were reported. A *p*-value ≤ 0.05 was considered statistically significant.

Sensitivity analysis was performed firstly considering GAD-2 and ISI scores as continuous variables in three series of multiple linear regressions involving each independent variable corrected, respectively, by sex and age (model 1); sex, age, and economic status (model 2); and sex, age, physical activity, and alcohol consumption (model 3). Secondly, for ISI score, the distribution of the independent variables was assessed across an alternative classification of severity of insomnia symptoms comprehensive of all the categories of insomnia symptom severity: no insomnia (ISI < 8), subclinical insomnia (8 ≤ ISI < 15), moderate insomnia (15 ≤ ISI < 22), and severe insomnia (ISI ≥ 22).

A moderation analysis was performed based on gender, using females as the baseline, employing multivariate logistic regressions with age adjustments for each independent variable and separately for the two main outcomes (GAD-2 ≥ 3 and ISI ≥ 15). An interaction term was generated, and significant effects were detected when both the predictor and the moderator significantly influenced the outcome. In instances where the interaction term was not significant, the predictor odds ratio (OR) is applicable to both males and females. However, in cases of significance, the interaction term applies to males, while the predictor relates to females.

All the analyses were performed using R 4.2.2 [48].

### 2.5. Ethical Considerations

Approval from an ethics committee was not necessary for this study, as the questionnaire responses were totally anonymous, preventing the identification of any participant, in accordance with European and Italian legislation [49,50,51]. Furthermore, the participation was voluntary and no incentive was used. Before accessing the questionnaire, residents had to declare that they understand the study’s methods and objectives providing explicit consent for their personal data processing. Although participant identification through questionnaire responses was not possible, data were downloaded and securely stored with a password after the collection period.

## 3. Results

### 3.1. Description of the Population

The characteristics of the sample are summarized in Table 1. The PHRASI study collected responses from 379 PHRs. Among these, the majority (58%, n = 219) were females, and the mean age was 30. For variables included in the objective workload sub-group, most of the sample was not employed in any additional job during their residency (64%, n = 242). The mean weekly working hours of the total sample were 38 h per week. A total of 304 residents (80%) answered that work duties did not interfere with private life. Moreover, 15% of respondents (n = 58) declared attending simultaneous traineeships in two or more units. Concerning variables included in the perceived workload sub-group, 15% of the sample (n = 55) reported that their workload was excessive. The mean score for the Work-SoC Manageability subscale was 4. Finally, for mental health outcomes, considering GAD-2, the mean score was 2.5 (SD = 1.7), and 36% (n = 137) were found to have symptoms of anxiety (GAD-2 ≥ 3). Regarding insomnia, the mean score was 7.7 (SD = 5.3), and 12% (n = 42) had a score above 14, the cut-off used to detect moderate-to-severe insomnia symptoms.

Table 2 shows the distribution of the independent variables across severity levels of anxiety and insomnia symptoms. Residents with anxiety symptoms (GAD-2 ≥ 3) perceived a higher workload and had lower scores of the Work-SoC Manageability subscale. In addition, their working hours for additional work contracts were more frequently considered valid for the completion of the residency postgraduate course and they were more frequently prone to report that their work duties interfered with their private life.

Considering the insomnia symptoms, residents with moderate-to-severe symptoms (ISI ≥ 15) were more frequently males; they were also more prone to attend simultaneous traineeships in more than one unit and experienced a higher work–life interference. Concerning the perceived workload variables, they reported a high perceived workload, and they feel less able to cope with their high workload.

### 3.2. Assessment of Collinearity

No significant collinearity was found among the variables (Figure 2). The strongest correlations were found between weekly working hours and having additional employment compatible with the postgraduate school (r = 0.65). The variable work–life interference was slightly correlated with a perceived high workload (r = 0.33) and inversely correlated with Work-SoC Manageability (r = −0.36). Insomnia symptoms were slightly correlated with anxiety symptoms (r = 0.42)

### 3.3. Main Analysis

A high perceived workload and more frequent work–life interference were associated with anxiety symptoms (GAD-2 ≥ 3) [aOR = 2.808, 95% CI = (1.559; 5.059), *p*-value = <0.001 and aOR = 2.937, 95% CI = (1.734; 4.974), *p*-value = <0.001], in the multiple logistic regression, adjusted for age and sex (model 1). Conversely, a higher score on the manageability subscale of Work-SoC was associated with lower odds of having anxiety symptoms [aOR = 0.639, 95% CI = (0.538; 0.760), *p*-value < 0.001]. The results do not significantly change after a further adjustment for economic status (model 2). Lastly, having an additional contract with working hours valid for completing the residency program resulted in this being significantly associated with lower odds of anxiety symptoms when further adjusting for sex, age, physical activity, and alcohol consumption (model 3) [aOR = 0.454, 95% CI = (0.208; 0.990), *p*-value = 0.047]. The results are reported in Table 3.

When considering insomnia symptoms, the multiple logistic regression, adjusted for age and sex (model 1), reveals that frequent work–life interference [aOR = 3.385, 95% CI = (1.689; 6.784), *p*-value = 0.001], the simultaneous attendance of two traineeships in different units [aOR = 2.783, 95% CI = (1.332; 5.814), *p*-value = 0.006], and a perceived high workload [aOR = 2.703, 95% CI = (1.239; 5.896), *p*-value = 0.012] were associated with moderate to severe insomnia symptoms. Moreover, a higher score on the manageability subscale of Work-SoC was inversely associated with insomnia symptoms [aOR = 0.645, 95% CI = (0.506; 0.821), *p*-value < 0.001]. Similar results were also found when further adjusting for economic status (model 2) and for physical activity and alcohol abuse (model 3). The results of the logistic regressions for insomnia symptoms (ISI ≥ 15) are shown in Table 4.

### 3.4. Sensitivity Analysis

When compared to the results of the logistic regression regarding GAD-2 with a cut-off of 3, the sensitivity analysis for anxiety symptoms shows more associations. In addition to the frequent work–life interference [aβ = 1.035, 95% CI (0.626; 1.445), *p* < 0.001], high perceived workload [aβ = 0.806, 95% CI (0.335; 1.277), *p* < 0.001], and low Work-SoC manageability [aβ = −0.415, 95% CI (−0.536; −0.295), *p* < 0.001] already highlighted in the main analysis, attending simultaneous traineeships also shows a statistically significant positive association with these symptoms [aβ = 0.483, 95% CI (0.019; 0.947), *p* = 0.041]. Conversely, compared to the main analysis, when performing the linear regressions, an opposite and negative relationship emerged between having an additional contract with working hours valid for completing the residency program and anxiety symptoms compared to having an additional contract with working hours not valid for the residency program [aβ = −0.740, 95% CI (−1.330; −0.150), *p* = 0.014]. The complete results of the sensitivity analysis for anxiety symptoms are visible in Appendix A.

The results of the sensitivity analysis for insomnia symptoms (visible in Appendix A) are entirely consistent with the main analysis: a high workload perceived [aβ = 1.640, 95% CI (0.069; 3.210), *p* = 0.041], frequent work–life interference [aβ = 2.204, 95% CI (0.844; 3.563), *p* = 0.001], and attending simultaneous traineeships [aβ = 2.459, 95% CI (0.973; 3.946), *p* = 0.001] remain positively associated with insomnia symptoms, while a high Work-SoC manageability still remains associated with lower levels of insomnia symptoms [aβ = −0.859, 95% CI (−1.272; −0.447), *p* < 0.001].

### 3.5. Moderation Analysis

The moderation analysis indicated the absence of a moderating effect of gender for all independent variables and for both the considered outcomes. Specifically, as reported in Appendix A, none of the interaction terms exhibited a statistically significant odds ratio. The interaction effect between individual predictors and sex (as a moderator) is shown in Appendix A. The figures support the conclusions drawn from the regressions.

Figure 3 presents a revised model of the expected relationships among the domains considered, based on the analysis results. Specifically, while certain aspects of the objective workload and all variables representing perceived workload were associated with both anxiety and insomnia symptoms, these associations were not influenced by sex in our sample.

## 4. Discussion

### 4.1. Interpretation of the Results

This study investigated workload characteristics and anxiety/insomnia symptoms among Italian PHRs during COVID-19, revealing significant prevalence rates: 36% reported anxiety, and 12% reported moderate-to-severe insomnia. Both objective workload (e.g., simultaneous traineeships and work–life interference) and subjective perceptions (e.g., high workload and manageability) consistently correlated with negative mental health outcomes. Objective factors like simultaneous traineeships and work–life interference, along with perceived workload such as a high workload and lack of manageability, were associated with anxiety and insomnia symptoms. These findings underscore the psychological burden among PHRs and highlight the importance of addressing workload factors to safeguard their mental well-being. Anxiety and insomnia are prevalent mental health issues among healthcare professionals, with rates often exceeding those in the general population. In our study, we observed that 36% of public health medical residents reported symptoms of anxiety, and 12% reported moderate-to-severe insomnia symptoms. These rates are comparable to findings in healthcare workers reported during the COVID-19 pandemic, where anxiety prevalence ranged from 26.3% to 31.8% and insomnia prevalence from 30.6% to 36.8% in healthcare populations [6,23,52]. However, compared to clinical physicians, the prevalence of insomnia symptoms in our sample is lower, which may reflect differences in the work environments and stress levels specific to public health versus clinical settings. Several factors may contribute to these differences. First, the nature of responsibilities and workload intensity can vary significantly between clinical and public health roles, with clinical positions often involving direct patient care, night shifts, and higher immediate physical and emotional demands. Conversely, public health residents may experience more variability in their workload, particularly with the dual commitments to academic and field-based tasks, which could differently affect mental health outcomes. Additionally, our sample primarily consisted of public health residents trained in preventive roles, which may provide different resources or stress management training compared to clinical settings.

Moreover, the timing of the data collection during the COVID-19 pandemic may partially explain elevated levels of anxiety and insomnia symptoms across studies. The pandemic added unprecedented demands on healthcare workers, with increased workload, exposure risks, and psychological stress. This contextual factor has been shown to exacerbate mental health issues, as noted in a meta-analysis, which found significantly higher rates of anxiety and insomnia in healthcare workers directly impacted by COVID-19 response roles.

Italian medical residents face a unique challenge balancing their education with traineeships of varying responsibilities, often opting for simultaneous traineeships to expedite skill development. However, managing tasks across different units with overlapping deadlines demands intense focus and dedication, potentially harming mental health in the long run. This overlap is a recognized issue, annually assessed by the Italian Observatory for Specializations Schools [53]. In 2022, 68.1% of public health medical residents reported an overlap, with 32.6% exceeding standard working hours. Despite this, simultaneous traineeships do not yield a higher income, unlike additional employment concurrent with residency, thus intensifying the adverse impact on mental health [54].

Considering the perceived workload, the results of the analyses confirm a strong association between both the two variables included for this dimension and all the mental health outcomes. In particular, the association of manageability, a dimension of work-sense of coherence, with these mental health symptoms means that the perception of having all the necessary resources to face work challenges plays an important role in residents’ psychological status. This is in line with the results of previous studies, in which a higher sense of coherence emerged as being negatively correlated with stress and depression and positively correlated with job satisfaction, well-being, and quality of life [55].

Some objective workload variables were not significantly associated with mental health outcomes, despite expectations post-COVID-19 that additional working hours and responsibilities among medical residents would exacerbate mental health risks [56]. Yet, these jobs provided crucial economic support, potentially mitigating their impact on mental health. Improved socioeconomic status, linked to better mental health outcomes, might explain why PHRs with extra employment or longer working hours did not show more anxiety or insomnia symptoms compared to those without [54]. Stratifying by economic status could verify this hypothesis, but the study’s low sample size precluded such an analysis. Nonetheless, economic status was a covariate in regression models, with no significant effect on the association between objective workload variables and mental health outcomes.

The study highlights a lack of association between certain objective workload variables (e.g., weekly working hours or having compatible additional employment) and mental health outcomes among PHRs, prompting a nuanced understanding. Unlike studies focusing on clinical doctors, the distinct attributes of public health professionals warrant a cautious interpretation of the findings [57]. While these variables measure the quantitative workload, they overlook the intensity, particularly during the pandemic, characterized by high work rates and urgent demands [58]. Work–life interference emerges as a more relevant indicator of mental well-being, reflecting how demanding schedules impact residents. Specific job demands, like a high workload, may indirectly affect mental health through emotional exhaustion or work–home interference [59]. Multicollinearity analysis suggests a correlation between working long hours and work–life interference, hinting at a potential cause-and-effect chain, with a high workload or additional work during residency leading to increased work–life interference, subsequently affecting mental health. This underscores the complex interplay between workload factors and mental well-being among PHRs.

Finally, the moderation analysis showed no gender differences in the association between workload characteristics and anxiety and insomnia symptoms, contrary to some existing evidence [60,61,62,63]. Further analyses are needed to confirm these results, but the study suggests that workload characteristics equally impact males and females regarding mental health outcomes.

### 4.2. Limitations and Strengths

Before generalizing the findings, certain considerations should be acknowledged. While the GAD-2 and ISI questionnaires are reliable for screening anxiety and insomnia symptoms, they do not substitute clinical diagnosis. However, their widespread use suggests a low likelihood of misdiagnosis [64,65]. Self-reported measures introduce recall and social desirability biases, despite anonymity [66]. This could be particularly true concerning sensitive information such as mental health symptoms. Future studies might consider incorporating objective measures or corroborating self-reports with clinical assessments to validate the accuracy of the reported symptoms and workload perceptions. Nevertheless, rigorous statistical methods, including sensitivity analyses, were employed to mitigate biases. Since both independent and dependent variables were reported by the participant, the common method bias could have influenced our results. Specifically, the social bias is subtle and difficult to mitigate, and, therefore, some response bias can be expected. Nevertheless, in order to address this, a series of effective and straightforward precautions were implemented. These included the provision of clear instructions before the completion of the survey, such as communicating that there were no correct answers; the assurance of the anonymity of the questionnaire, with online administration and no contact with the interviewer; the avoidance of complex and ambiguous items; the separation of the section containing the questions dedicated to measuring the independent variables from the one containing the questions measuring the dependent variables; and the relative conciseness of the questionnaire. All these techniques have been documented to be effective in reducing the common method bias [67,68,69,70]. The cross-sectional design limits establishing cause-and-effect relationships. Yet, given the lack of prior research, the survey was a practical approach. Moreover, the cross-sectional nature of this study limits the ability to infer causation between workload characteristics and mental health outcomes. While our findings suggest associations, longitudinal studies are necessary in order to establish causal relationships and to understand how workload factors influence mental health over time. Future studies could employ a longitudinal design to track changes in workload and mental health status, providing deeper insights into causal pathways. Lastly, we did not perform a confirmatory factor analysis (CFA) for each measurement scale used, such as the Generalized Anxiety Disorder-2 (GAD-2), the Insomnia Severity Index (ISI), and the Work-Related Stress Questionnaire (WRSQ). While these instruments have been previously validated in similar populations, including our own sample for the WRSQ (as indicated by the reported Cronbach’s alpha), the absence of CFA represents a limitation in terms of scale structure verification within the context of our specific study population. The primary reason for not conducting a CFA was due to the study’s focus on exploring associations between workload characteristics and mental health outcomes, rather than on validating the psychometric properties of each scale. Furthermore, resource constraints, particularly related to the sample size and the scope of the analysis, limited our capacity to include a CFA. We recommend that future research building on our findings consider a more in-depth psychometric validation, including CFA, to strengthen the reliability of results in similar contexts.

Despite these limitations, the study’s strengths include the use of validated scales for the mental health assessment, and a comprehensive approach to examining both objective and perceived workload characteristics; and a comprehensive coverage of the reference population and a representative sample, with nationwide involvement as described in a previous work [71]. The substantial sample size exceeded the minimum requirement, and measures were taken to ensure questionnaire completion by the targeted population [30,72]. Efficiently conducted with minimal cost and within a short timeframe, the research maintained complete data. The study’s robustness is further reinforced by thorough statistical analyses, including assessing multicollinearity and conducting sensitivity analyses using various classification systems for questionnaire scores. Furthermore, the moderation analysis enhances the breadth of the study by accounting for gender differences, a crucial aspect in mental health research, thus providing a more comprehensive perspective. Our findings provide a foundation for future work aiming to enhance residency training environments and support the mental health of public health residents.

### 4.3. Implication for Practice, Policy, and Research

This study offers significant theoretical contributions to the understanding of how workload characteristics influence mental health outcomes, specifically, anxiety and insomnia symptoms, among public health medical residents. Theoretically, our findings expand the literature on occupational stress and mental health within healthcare settings, focusing on the unique context of residency programs. Unlike other healthcare professionals, residents are in a dual role, balancing academic requirements and clinical responsibilities, which introduces distinct stressors that impact their well-being. This duality contributes to a better understanding of the psychological challenges within medical training environments, particularly in public health fields. Our study supports the theory that perceived workload and specific job demands—such as work–life interference and simultaneous traineeships—are critical determinants of mental health outcomes. These results align with and extend occupational health models by highlighting how these workload factors differentially impact this specific population.

Our findings also contribute to a broader understanding of occupational stress theory by emphasizing the role of perceived manageability within highly structured training environments such as medical residencies. This study suggests that perceived workload and manageability are not simply reflections of the number of tasks to be managed, but are fundamental factors influencing psychological well-being and mental health. By placing these factors in the specific, high-risk context of public health specialization, our research highlights the need to refine theoretical models of work stress to account for the dual pressures, both academic and clinical, that uniquely impact this population. This insight enriches the conceptual frameworks that guide the design of residency programs, suggesting the importance of incorporating workload management and resilience-building strategies at a structural level into healthcare training systems.

By demonstrating a direct correlation between a high workload among PHRs and anxiety and insomnia, consistent with the previous research that highlighted the stressful nature of residency training for medical doctors, our findings emphasize the necessity of shaping a safe and healthy work environment to enhance residents’ mental health [73,74].

The COVID-19 pandemic has exacerbated mental health challenges, notably among healthcare professionals [52]. Depression and anxiety, common disorders, significantly impact quality of life and cognitive abilities [75]. Introducing mental health screening and support in PHR workplaces is crucial. Despite their elevated risk, PHRs often underutilize mental health services due to various barriers [76,77]. Thus, ensuring accessible support is imperative. Additionally, systematically addressing workload factors linked to mental health symptoms, like strengthening supervision, is vital to enhancing PHRs’ training environment and sense of manageability. Recent analyses suggest supervisor support may offer a protective role [78,79]. Overcoming the stigma surrounding mental health requires a multifaceted approach involving education, awareness campaigns, and fostering inclusive work environments [80,81]. In practical terms, the findings have implications for improving residency programs and supporting the mental well-being of public health residents. The identification of specific workload stressors, such as frequent work–life interference and the need to manage multiple traineeships, suggests areas where program structures could be modified. Residency program directors and policymakers could consider limiting overlapping clinical duties or providing structured support resources, such as mentorship programs and mental health services, tailored to the needs of residents. Additionally, enhancing supervisor support and fostering a supportive work environment may mitigate the adverse effects of high workload demands, as our findings indicate that workload manageability correlates positively with mental health outcomes. Addressing these factors could reduce burnout, increase job satisfaction, and improve both the mental health and performance of residents.

These theoretical insights and practical recommendations underscore the importance of comprehensive support systems within residency programs, aimed at fostering resilience and well-being among residents. Future research should explore the longitudinal impacts of workload adjustments to better understand causality and long-term benefits for mental health among public health medical residents. Particularly, our study opens new perspectives for future research to delve into various facets of how workload affects the mental health of medical residents. Indeed, further investigations are warranted to examine how socioeconomic status potentially moderates the effects of extended work hours and additional employments on mental well-being. Furthermore, considering the substantial body of evidence on gender differences in mental health challenges, future research could apply gender-specific analyses to confirm our results. Addressing these queries may necessitate a longitudinal approach, which could provide an added temporal dimension for assessing causality in the relationships elucidated in this study. For this reason, the “Public Mental Health” working group of the medical residents’ Assembly of the Italian Society of Hygiene, Preventive Medicine, and Public Health has designed the “Residents’ Mental Health Investigation, a Dynamic Longitudinal Study in Italy” (ReMInDIt) [82]. Currently in the data collection phase, this study will contribute in shedding light on the questions emerging from the PHRASI study.

## 5. Conclusions

The present analysis from the PHRASI study offers crucial insights into mental health symptoms among Italian PHRs, producing the first nation-wide estimates of anxiety and insomnia symptom prevalence. Moreover, it focuses on their workload characteristics, describing them from both an objective and perceived point of view. Additionally, it pinpoints those workload characteristics that are linked to the presence of anxiety and insomnia symptoms, such as frequent work–life interference, the attendance of simultaneous traineeships in different units, a high perceived workload, and the absence of sufficient internal and external resources to face job demands. This newfound awareness may prompt prospective interventions in Public Health residencies to diminish the occurrence of overwork among PHRs, enhance the work–life balance, and advocate for a healthier work environment.

## Figures and Tables

**Figure 1 healthcare-12-02299-f001:**
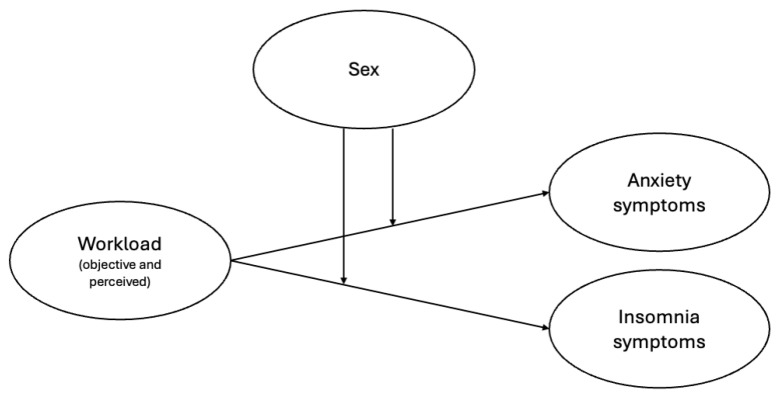
Graphical representation of Hypotheses 1 and 2, illustrating the expected relationships among the domains analyzed in study.

**Figure 2 healthcare-12-02299-f002:**
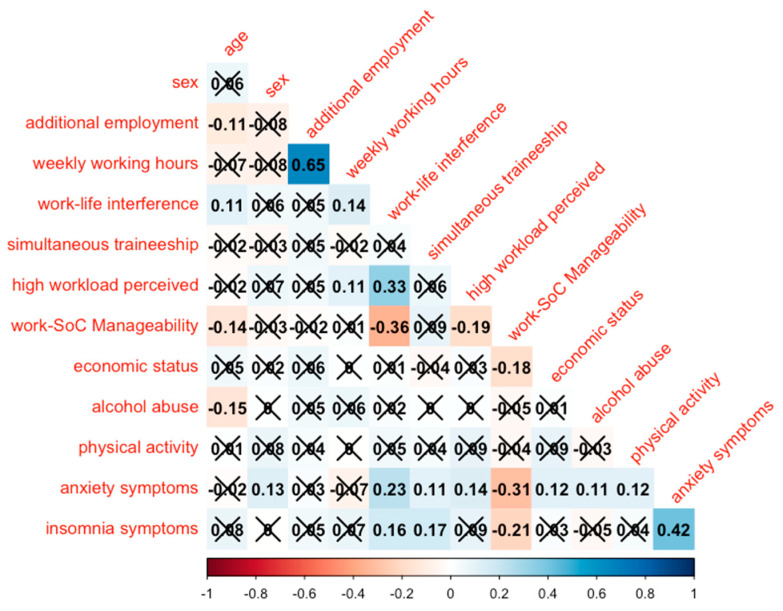
Correlation matrix of the variables under study. The values marked with an X were not statistically significant.

**Figure 3 healthcare-12-02299-f003:**
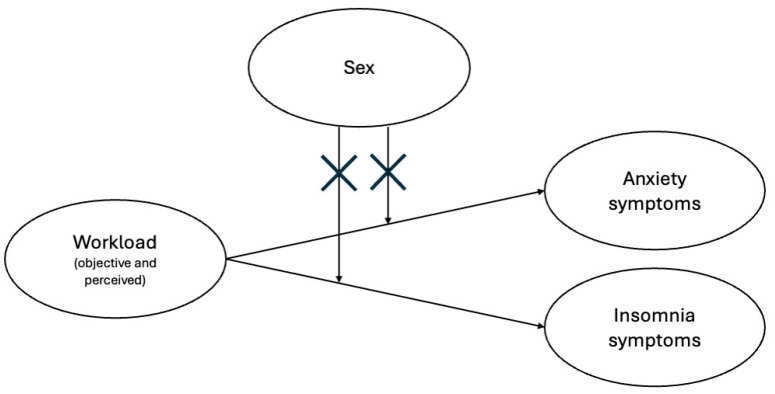
Graphical representation of the relationships among the domains analyzed in the study. The “X” on the arrows means the absence of a moderating effect of sex.

**Table 1 healthcare-12-02299-t001:** Distribution of selected characteristics of the study population.

Characteristic	N = 379
*Demographic data*	
Age (years)	30.0 (29.0, 34.0) ^1^
Sex	
Female	219 (58%)
Male	160 (42%)
*Objective workload*	
Additional employment	
No	242 (64%)
Yes	137 (36%)
Additional contract’s working hours valid for residency completion	
No	88 (68%)
Yes	42 (32%)
Weekly working hours	38 (38, 38)
Simultaneous traineeships	
No	321 (85%)
Yes	58 (15%)
Work–life interference	
Infrequent	304 (80%)
Frequent	75 (20%)
*Perceived workload*	
High workload perceived	
No	324 (85%)
Yes	55 (15%)
Work-SoC Manageability	4.00 (3.00, 5.00) ^1^
*Mental health*	
Anxiety symptoms (GAD-2)—*continuous scale*	2.5 (1.7) ^2^
Anxiety symptoms (GAD-2)	
GAD-2 < 3	242 (64%)
GAD-2 ≥ 3	137 (36%)
Insomnia symptoms (ISI)—*continuous scale*	7.7 (5.3) ^2^
Insomnia symptoms (ISI)	
Subclinical or absent insomnia (ISI < 15)	323 (88%)
Moderate to severe (ISI ≥ 15)	42 (12%)

^1^ Median (IQR); ^2^ Mean (SD).

**Table 2 healthcare-12-02299-t002:** Relations between sample characteristics and mental health symptoms.

Characteristic	GAD-2 < 3, N = 242 ^1^	GAD-2 ≥ 3, N = 137 ^1^	*p*-Value	ISI < 15, N = 323 ^1^	ISI ≥ 15, N = 42 ^1^	*p*-Value
*General data*						
Age	31.0 (28.0, 34.0)	30.0 (29.0, 33.0)	0.5 ^2^	30.0 (29.0, 34.0)	31.5 (30.0, 34.0)	0.11 ^2^
Sex			0.090 ^3^			**0.037** ^3^
Female	132 (55%)	87 (64%)		193 (60%)	18 (43%)	
Male	110 (45%)	50 (36%)		130 (40%)	24 (57%)	
*Objective workload*						
Additional employment			0.6 ^3^			0.2 ^3^
No	157 (65%)	85 (62%)		210 (65%)	23 (55%)	
Yes	85 (35%)	52 (38%)		113 (35%)	19 (45%)	
Additional contract’s working hours valid for residency completion			**0.045** ^3^			0.8 ^3^
No	60 (74%)	28 (57%)		73 (67%)	12 (71%)	
Yes	21 (26%)	21 (43%)		36 (33%)	5 (29%)	
Weekly working hours	38 (38, 38)	38 (38, 38)	0.33	38 (38, 38)	38 (38, 47)	0.3 ^3^
Simultaneous traineeships			0.4 ^3^			**0.004** ^3^
No	208 (86%)	113 (82%)		279 (86%)	29 (69%)	
Yes	34 (14%)	24 (18%)		44 (14%)	13 (31%)	
Work–life interference			**<0.001** ^3^			**<0.001** ^3^
Infrequent	95 (69%)	209 (86%)		268 (83%)	25 (60%)	
Frequent	42 (31%)	33 (14%)		55 (17%)	17 (40%)	
*Perceived workload*						
High workload perceived			**<0.001** ^3^			**0.015** ^3^
No	219 (90%)	105 (77%)		283 (88%)	31 (74%)	
Yes	23 (9.5%)	32 (23%)		40 (12%)	11 (26%)	
Work-SoC Manageability	4.50 (3.50, 5.00)	3.50 (2.50, 4.50)	**<0.001** ^2^	4.00 (3.50, 5.00)	3.50 (2.50, 4.00)	**<0.001** ^2^

Statistically significant *p*-values are in bold. ^1^ Median (IQR); n (%). ^2^ Wilcoxon–Mann–Whitney test; ^3^ Pearson’s Chi-squared test.

**Table 3 healthcare-12-02299-t003:** Results of multivariate logistic regression analysis for anxiety symptoms (GAD-2 ≥ 3).

	Model 1 ^a^			Model 2 ^b^			Model 3 ^c^		
Characteristic	aOR	95% CI	*p*-Value	aOR	95% CI	*p*-Value	aOR	95% CI	*p*-Value
*Objective workload*									
Additional employment (Ref. No)									
Yes	1.146	0.738–1.781	0.543	1.110	0.712–1.730	0.645	1.100	0.704–1.718	0.676
Additional contract’s working hours valid for residency completion (Ref. No)									
Yes	0.475	0.221–1.020	0.056	0.471	0.218–1.015	0.055	0.454	0.208–0.990	**0.047**
Weekly working hours	0.985	0.959–1.011	0.255	0.984	0.958–1.011	0.238	0.982	0.956–1.009	0.198
Simultaneous traineeships (Ref. No)									
Yes	1.325	0.746–2.355	0.337	1.352	0.759–2.409	0.306	1.270	0.710–2.273	0.42
Work–life interference (Ref. Infrequent)									
Frequent	2.937	1.734–4.974	**<0.001**	2.972	1.749–5.049	**<0.001**	2.847	1.672–4.847	**<0.001**
*Perceived workload*									
High workload perceived (Ref. No)									
Yes	2.808	1.559–5.059	**<0.001**	2.777	1.539–5.009	**0.001**	2.728	1.505–4.947	**0.001**
Work-SoC Manageability	0.639	0.538–0.760	**<0.001**	0.646	0.542–0.770	**<0.001**	0.647	0.543–0.770	**<0.001**

Statistically significant *p*-values are in bold. ^a^ Adjusted for sex and age; ^b^ Adjusted for sex, age, and economic status; ^c^ Adjusted for sex, age, physical activity, and alcohol abuse.

**Table 4 healthcare-12-02299-t004:** Results of multivariate logistic regression analysis for moderate-to-severe insomnia symptoms (ISI ≥ 15).

	Model 1 ^a^			Model 2 ^b^			Model 3 ^c^		
Characteristic	aOR	95% CI	*p*-Value	aOR	95% CI	*p*-Value	aOR	95% CI	*p*-Value
*Objective workload*									
Additional employment (Ref. No)									
Yes	1.521	0.786–2.945	0.213	1.488	0.767–2.888	0.240	1.511	0.776–2.939	0.224
Additional contract’s working hours valid for residency completion (Ref. No)									
Yes	0.960	0.305–3.020	0.945	1.030	0.327–3.248	0.960	1.153	0.356–3.728	0.812
Weekly working hours	1.016	0.981–1.052	0.371	1.016	0.981–1.052	0.375	1.015	0.981–1.052	0.39
Simultaneous traineeships (Ref. No)									
Yes	2.783	1.332–5.814	**0.006**	2.822	1.347–5.913	**0.006**	2.756	1.314–5.777	**0.007**
Work–life interference (Ref. Infrequent)									
Frequent	3.385	1.689–6.784	**0.001**	3.424	1.705–6.878	**0.001**	3.342	1.661–6.724	**0.001**
*Perceived workload*									
High workload perceived (Ref. No)									
Yes	2.703	1.239–5.896	**0.012**	2.663	1.218–5.822	**0.014**	2.762	1.260–6.055	**0.011**
Work-SoC Manageability	0.645	0.506–0.821	**<0.001**	0.649	0.507–0.831	**0.001**	0.647	0.507–0.825	**<0.001**

Statistically significant *p*-values are in bold. ^a^ Adjusted for sex and age; ^b^ Adjusted for sex, age, and economic status; ^c^ Adjusted for sex, age, physical activity, and alcohol abuse.

## Data Availability

The authors can be contacted for information about the data presented.

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
