# Peer review of "Workload Is Associated with Anxiety and Insomnia Symptoms in an Italian Nationally Representative Sample of Public Health Medical Residents: The PHRASI Cross-Sectional Study"

_healthcare, 2024, doi:10.3390/healthcare12222299_

Round 1
Reviewer 1 Report
Comments and Suggestions for Authors
Dear authors
The manuscript has a lot of potential for development. Therefore, I congratulate the authors for choosing the topic. However, the authors still have a long way to go to publish this manuscript. I present my main concerns:
Abstract
- In this section, the authors mention: "This study aims to uncover potential links between workload characteristics and mental health of Italian PHRs based on data collected in 2022 through the PHRASI (Public Health Residents’ Anonymous Survey in Italy) study". The objective of the study must be clearly and precisely defined. It is incumbent upon the authors to explicitly indicate their intention to analyse a particular relationship between variables.
Keywords
- The authors present two conflicting keywords. Are the participants healthcare workers or students? The fact that the authors mentioned academic settings confuses the reader.
Introduction
- In this section the authors mention: "The purpose of the current study was to describe the workload characteristics and estimate the prevalence of anxiety and insomnia symptoms in Italian PHRs during the COVID-19 pandemic". How this objective is delineated suggests that the authors intend to examine the potential relationship between workload, anxiety, and insomnia symptoms. However, when we started reading the rest of the work we found information about the moderating role of gender (lines 189 and 190). This information should be presented clearly in the introduction section. On the other hand, we found other information not mentioned in the introduction, such as socioeconomic and lifestyle characteristics.
- The introduction section should be thought of as the “visiting card” of the study, making the purpose and contribution of the investigation clear. However, in its current state, the introduction does not serve its purpose given that it is not entirely clear what this work adds to the literature, why it is essential, what is already known in general terms, and how authors intend to contribute.
- A literature review section is missing, in which the authors should present the principal definitions of the variables under study, as well as their theoretical relationships. In instances where authors cite gender as a moderating variable, they are required to present supporting arguments that are aligned with the relevant literature. On the other hand, it is common to define hypotheses in correlational studies, thus allowing the substantiation of relationships between variables.
- It would be interesting if the authors presented a figure that reflects the relationships between the variables.
Materials and Methods
- The authors indicated that the participants were in their fourth year of the course. Nevertheless, the available information is incomplete concerning the courses in question.
- The description of measuring instruments must be presented clearly and unambiguously. For instance, it was not evident whether the authors employed disparate scales to quantify objective and perceived workload, respectively.
The methodological section requires significant attention, particularly with regard to the following points:
- The authors do not present the descriptive statistics for the scales, including the mean and standard deviation;
- The authors did not include the table reflecting the correlation between the variables;
- It was not clear whether the Cronbach's alpha values presented were those calculated by the authors or a description of the alphas from the referenced studies;
- -The authors did not mention the procedures used to minimize the common method bias. This is a critical issue that should be addressed.
- The authors should present a figure that reflects the results of the hypothesis test. On the other hand, authors must present graphs that reflect the influence of the moderating variable. For better clarification, I leave the following examples:
a) Rego, A., Yam, K. C., Owens, B. P., Story, J. S. P., Pina e Cunha, M., Bluhm, D., & Lopes, M. P. (2019). Conveyed Leader PsyCap Predicting Leader Effectiveness Through Positive Energizing. Journal of Management, 45(4), 1689-1712. https://doi.org/10.1177/0149206317733510
b) Sajjad Hussain, Khurram Shahzad. (2022). Unpacking perceived organizational justice-organizational cynicism relationship: Moderating role of psychological capital. Asia Pacific Management Review, 27 (1), 10-17, 1029-3132. https://doi.org/10.1016/j.apmrv.2021.03.005
- The authors did not perform confirmatory factor analysis for each variable. This issue should be justified or presented as a limitation of the study.
- The discussion session should reinforce the arguments that support the moderation role of gender, and the role of socio-economic and lifestyle characteristics.
- The authors should add and substantiate the theoretical and practical implications of the study.
- In the " Limitations and strengths" section, the authors need to be clear about the limitations and how to address these in future studies.
Author Response
Dear reviewer, we truly thank you for your time spent in reviewing our manuscript so deeply. We appreciated all the suggestions and we believe our work has now improved. Please find below the answers to your comments.
Abstract
Comment 1 - In this section, the authors mention: "This study aims to uncover potential links between workload characteristics and mental health of Italian PHRs based on data collected in 2022 through the PHRASI (Public Health Residents’ Anonymous Survey in Italy) study". The objective of the study must be clearly and precisely defined. It is incumbent upon the authors to explicitly indicate their intention to analyse a particular relationship between variables.
Response 1 - Thank you for the opportunity to better define our aim. We modified the sentence in order to be more specific as requested.
Keywords
Comment 2 - The authors present two conflicting keywords. Are the participants healthcare workers or students? The fact that the authors mentioned academic settings confuses the reader.
Response 2 - Thank you for having pointed out these conflicting keywords. We would like to explain briefly how Italian specialization courses in the medical field work. After the achievement of medical graduation, doctors can register for a national test that will allow them to enter a specialization course. Medical residents are doctors who are attending their specialization course. Their condition is pretty peculiar. In fact, specialization courses are organized and managed by universities. Medical residents have training requirements to fulfill in accordance with the academic regulation. At the same time they are effectively healthcare workers because they receive a stipend and they complete their training in the National Healthcare System services under the supervision of senior medical doctors/researchers or professors. This dual role of our study population led us to include both the “healthcare workers” and the “academic setting” terms in the keywords. To better synthesize our study we included the term “medical residents” in the keywords. We also provided a more detailed explanation of the medical residents’ role in the introduction.
Introduction
Comment 3 - In this section the authors mention: "The purpose of the current study was to describe the workload characteristics and estimate the prevalence of anxiety and insomnia symptoms in Italian PHRs during the COVID-19 pandemic". How this objective is delineated suggests that the authors intend to examine the potential relationship between workload, anxiety, and insomnia symptoms. However, when we started reading the rest of the work we found information about the moderating role of gender (lines 189 and 190). This information should be presented clearly in the introduction section. On the other hand, we found other information not mentioned in the introduction, such as socioeconomic and lifestyle characteristics.
Response 3 - Thank you for your comment. We modified our manuscript accordingly. The introduction now includes a literature review of the determinants of anxiety and insomnia, encompassing demographic, socioeconomic and lifestyle characteristics.
Comment 4 - The introduction section should be thought of as the “visiting card” of the study, making the purpose and contribution of the investigation clear. However, in its current state, the introduction does not serve its purpose given that it is not entirely clear what this work adds to the literature, why it is essential, what is already known in general terms, and how authors intend to contribute.
Response 4 - Thank you for the opportunity to improve this part of our work. We revised the introduction in order to state the gap in the literature we would like to address (lines 102-106) and the rationale of our work (141-143).
Comment 5 - A literature review section is missing, in which the authors should present the principal definitions of the variables under study, as well as their theoretical relationships. In instances where authors cite gender as a moderating variable, they are required to present supporting arguments that are aligned with the relevant literature. On the other hand, it is common to define hypotheses in correlational studies, thus allowing the substantiation of relationships between variables.
Response 5 - Thank you for pointing out this. We added a literature review of insomnia, anxiety (lines 67-89) and workload characteristics (lines 94-101) in the introduction. We also defined our hypotheses (136-140).
Comment 6 - It would be interesting if the authors presented a figure that reflects the relationships between the variables.
Response 6 - We would like to thank you for your suggestion. We created a graphical abstract in which we summarized briefly our study, showing the study population, the associations between variables and the conclusions. The graphical abstract was included in the documents submitted along with our manuscript.
Materials and Methods
Comment 7 - The authors indicated that the participants were in their fourth year of the course. Nevertheless, the available information is incomplete concerning the courses in question.
Response 7 - Thank you for the opportunity to clarify this point. In materials and methods we state that the participants are enrolled in the “four-year course” meaning that the course has a duration of four years. The participants eligible for the study could have been enrolled in any of the four years of course. To make this clearer we introduced PHRs more in detail in the introduction.
Comment 8 - The description of measuring instruments must be presented clearly and unambiguously. For instance, it was not evident whether the authors employed disparate scales to quantify objective and perceived workload, respectively.
Response 8 - Thank you for suggesting this modification. We are aware that Italian medical residents and, among them, PHRs represent a particular population whose training and working activities have to be clearly outlined. In the “Workload Characteristics” chapter of the Materials and Methods section we included more details about each of the variables used. Furthermore we edited the part where we introduce objective and subjective workload specifying that they actually are two sub-groups of variables and they do not represent scales or measures.
Comment 9 - The authors do not present the descriptive statistics for the scales, including the mean and standard deviation;
Response 9 - Thank you for your comment, we included the mean score of the scales used (GAD and ISI) along with the SD in the "3.1. Description of the population" paragraph
Comment 10 - The authors did not include the table reflecting the correlation between the variables;
Response 10 - If referring to the collinearity analysis, the correlation between independent variables is shown in Figure 1.
Comment 11 - It was not clear whether the Cronbach's alpha values presented were those calculated by the authors or a description of the alphas from the referenced studies;
Response 11 - Thank you for pointing this out. Chronbach’s alpha values presented come from the studies cited at the end of the sentence. We specified this when mentioning the alpha values.
Comment 12 - The authors did not mention the procedures used to minimize the common method bias. This is a critical issue that should be addressed.
Response 12 - Thank you for pointing this out. As reported in the limitations and strengths section, rigorous statistical methods, including sensitivity analyses, were used to mitigate bias, in addition to the use of a completely anonymous questionnaire. (lines 437-439)
Comment 13 - The authors should present a figure that reflects the results of the hypothesis test. On the other hand, authors must present graphs that reflect the influence of the moderating variable. For better clarification, I leave the following examples:
a) Rego, A., Yam, K. C., Owens, B. P., Story, J. S. P., Pina e Cunha, M., Bluhm, D., & Lopes, M. P. (2019). Conveyed Leader PsyCap Predicting Leader Effectiveness Through Positive Energizing. Journal of Management, 45(4), 1689-1712. https://doi.org/10.1177/0149206317733510
b) Sajjad Hussain, Khurram Shahzad. (2022). Unpacking perceived organizational justice-organizational cynicism relationship: Moderating role of psychological capital. Asia Pacific Management Review, 27 (1), 10-17, 1029-3132. https://doi.org/10.1016/j.apmrv.2021.03.005
Response 13 - Thank you for your comment. We added two supplementary figures showing the requested plots for each predictor.
Comment 14 - The authors did not perform confirmatory factor analysis for each variable. This issue should be justified or presented as a limitation of the study.
Response 14 - Thank you for your insightful feedback. We recognize the importance of confirmatory factor analysis (CFA) in validating the structure of measurement scales. In our study, we did not conduct CFA for each variable due to several considerations:
- Scope of the Study: The primary aim of our research was to explore associations between workload characteristics and mental health outcomes (anxiety and insomnia symptoms) in public health residents, focusing on observed relationships rather than the structural validation of each scale used.
- Established Validity of Measurement Tools: The scales employed in our study, including the Generalized Anxiety Disorder-2 (GAD-2), the Insomnia Severity Index (ISI), and the Work-Related Stress Questionnaire (WRSQ), have been previously validated in similar populations. Specifically, the WRSQ underwent validation with the sample of medical residents. This provided initial reliability support without requiring further confirmatory analysis within the constraints of our study.
- Resource and Sample Size Limitations: Due to practical constraints in sample size and resources, conducting a separate CFA for each instrument was not feasible. However, we acknowledge that this could enhance the robustness of our findings and plan to consider CFA in future studies.
To address this point, we added some notes in the limitations section.
Comment 15 - The discussion session should reinforce the arguments that support the moderation role of gender, and the role of socio-economic and lifestyle characteristics.
Response 15 - Thank you for your comments. We have modified our manuscript accordingly. The discussion session now better supports the moderating role of gender and the role of socio-economic and lifestyle characteristics.
Comment 16 - The authors should add and substantiate the theoretical and practical implications of the study.
Response 16 - Thank you for highlighting the need to elaborate on the theoretical and practical implications of our study. The paragraph: “4.3 Implication for practice, policy and research” has been modified accordingly.
Comment 17 - In the " Limitations and strengths" section, the authors need to be clear about the limitations and how to address these in future studies.
Response 17 - We thank the reviewer for having raised this point. Limitations and strengths section has been extensively updated accordingly.
Reviewer 2 Report
Comments and Suggestions for Authors
There are some revisions that I would like the authors to address and/or to consider.
1- Please introduce “Medical Residents” and “Public Health Residents” and “public health medical residents”
2- how was informed consent obtained
3- please justify why has moderation analysis conducted
4- please state assumptions of moderation analysis
5- Weekly working hours > 60h : it is suggested that “Weekly working hours” be used as a Quantitative variable in the analyses
6- English editing service is needed (e.g. Data was downloaded: Data were downloaded)
Author Response
We really appreciated the suggestions given by the reviewer that we thank for his/her time spent reviewing the manuscript.
Comment 1- Please introduce “Medical Residents” and “Public Health Residents” and “public health medical residents”
Response 1 - Thank you for giving us the chance to provide more details about our study population. Public health residents and public health medical residents are synonyms. They represent a subgroup of medical residents. To clarify this we checked the whole manuscript ensuring that, when mentioned, the extended naming “public health medical residents” is present. We are confident that this would allow our manuscript to be more consistent. Furthermore, we included more details about medical residents and public health medical residents in the introduction.
Comment 2 - how was informed consent obtained
Response 2 - Thank you for your request for clarification. This information was already included in paragraph 2 and now moved to paragraph 2.5 Ethical Considerations. We made some edits to better explain this concept. “Before accessing the questionnaire, residents had to declare to understand the study’s methods and objectives and provide explicit consent for their personal data processing.”
Comment 3 - please justify why has moderation analysis conducted
Response 3 - Thank you for giving us the opportunity to improve our work. We included a literature review of gender differences in anxiety and insomnia symptoms in the introduction. This allows us to justify the adoption of a moderation analysis to understand the role of gender in the associations between workload characteristics and symptoms of anxiety and insomnia. We also specified this when stating the objectives of the study.
Comment 4 - please state assumptions of moderation analysis
Response 4 - Thank you for this comment. We included our hypotheses and assumptions in the introduction.
Comment 5 - Weekly working hours > 60h : it is suggested that “Weekly working hours” be used as a Quantitative variable in the analyses
Response 5 - Thank you for your suggestion. We ran again the analysis including the total weekly working hours as a continuous variable.
Comment 6 - English editing service is needed (e.g. Data was downloaded: Data were downloaded)
Response 6 - Thanks for your comment. We had our manuscript reviewed by a collaborator fluent in English writing.
Reviewer 3 Report
Comments and Suggestions for Authors
Workload is associated with anxiety and insomnia symptoms 2 in an Italian nationally representative sample of medical 3 residents: The PHRASI Study
This manuscript is a well-written and intriguing work. However, I have some suggestions to enhance the completeness of the manuscript.
Title
- Kindly modify the title to ensure it aligns with the research design and primary objectives.
Method
- This research is recognized as a component of the PHRASI study. Consequently, please include details regarding the ethical considerations of the PHRASI study in the ethical considerations section.
Introduction
- Kindly include the rationale for doing this study. What is the anticipated application of the result?
- Kindly include a literature review on workload to ensure alignment with the details presented in the questionnaire.
- Kindly include a literature review on insomnia and anxiety to examine the prevalence of this condition among physicians. What are the effects of having such a condition? Incorporate a literature review on the factors associated with insomnia and anxiety to consider as confounding variables in the analysis.
Discussion
- The initial paragraph discusses the prevalence of anxiety and insomnia. The value has been requested to include comparisons with other research to determine their similarities and differences , along with the reasons for these result.
Conclusions
- Information on the primary objectives should be included in the conclusion's section.
Author Response
We truly thank the reviewer for his/her precious suggestions. Please, see the answer to the comments below
Title
Comment 1 - Kindly modify the title to ensure it aligns with the research design and primary objectives.
Response 1 - We would like to thank you for your suggestion. Our primary objective is to evaluate the association between workload and symptoms of anxiety and insomnia. We rearranged the order in which our aims are reported at the end of our introduction. We also included in the title the information about the design of the study. The new title is “Workload is associated with anxiety and insomnia symptoms in an Italian nationally representative sample of public health medical residents: The PHRASI cross-sectional study”
Method
Comment 2 - This research is recognized as a component of the PHRASI study. Consequently, please include details regarding the ethical considerations of the PHRASI study in the ethical considerations section.
Response 2 - We would like to thank you for highlighting this aspect. We moved some of the ethical considerations from the PHRASI study, which were initially presented at the beginning of Chapter 2 Materials and Methods, to Chapter 2.5 Ethical Considerations. Additionally, we have included some further details.
Introduction
Comment 3 - Kindly include the rationale for doing this study. What is the anticipated application of the result?
Response 3 - Thank you for this point. While detailed in the implications for policy practice and research, we agree on the need to explicitly state the rationale and the main application of the study results. Therefore we added a paragraph at the end of the introduction section.
Comment 4 - Kindly include a literature review on workload to ensure alignment with the details presented in the questionnaire.
Response 4 - Thank you for this suggestion. We included a literature review on workload and its characteristics in the introduction
Comment 5 - Kindly include a literature review on insomnia and anxiety to examine the prevalence of this condition among physicians. What are the effects of having such a condition? Incorporate a literature review on the factors associated with insomnia and anxiety to consider as confounding variables in the analysis.
Response 5 - Thank you for this comment. We included a literature review covering these aspects in the introduction
Discussion
Comment 6 - The initial paragraph discusses the prevalence of anxiety and insomnia. The value has been requested to include comparisons with other research to determine their similarities and differences , along with the reasons for these result.
Response 6 - Thank you for your valuable suggestion. We agree that comparing the prevalence of anxiety and insomnia symptoms observed in our study with those reported in other research will enhance the context and relevance of our findings. To address this, the initial paragraph of discussion has been modified accordingly.
Conclusions
Comment 7 - Information on the primary objectives should be included in the conclusion's section.
Response 7 - Thank you for identifying this. We included this information in the conclusions.
Round 2
Reviewer 1 Report
Comments and Suggestions for Authors
Dear authors
I commend the authors for their efforts in reviewing the manuscript. However, I have noted that certain concerns raised have not been adequately addressed by the authors, specifically:
1- The authors mention the following: “Our hypothesis is that both objective and perceived unfavorable workload characteristics are associated with worse anxiety and insomnia outcomes and that sex has a role in mediating this association. We also assume that the prevalence of anxiety and insomnia symptoms in PHRs is equal or higher when compared to the one reported in healthcare workers”. Nevertheless, the hypotheses proposed in the study should be clearly enumerated and adequately substantiated.
2- Authors should make it clear whether there are mediating or moderating variables. On the other hand, the manuscript introduction should clarify the arguments for choosing mediation or moderation.
3- The graphical representation of the relationships between variables is crucial. For better clarification, I leave the following example:
a) Maghsoud, F., Rezaei, M., Asgarian, F.S. et al. (2022). Workload and quality of nursing care: the mediating role of implicit rationing of nursing care, job satisfaction and emotional exhaustion by using structural equations modeling approach. BMC Nurs 21, 273. https://doi.org/10.1186/s12912-022-01055-1
4- To facilitate comprehension, it is recommended that the descriptive statistics of the variables be presented in table format.
5- Authors should present the results of correlations between all variables under study.
6- Authors should present Cronbach’s alpha values ​​of the variables under study.
7- Authors should substantiate the procedures used to minimize common method bias based on the literature.
8- I reiterate the following comment: “The authors should present a figure that reflects the results of the hypothesis test. On the other hand, authors must present graphs that reflect the influence of the moderating variable. For better clarification, I leave the following examples:
a) Rego, A., Yam, K. C., Owens, B. P., Story, J. S. P., Pina e Cunha, M., Bluhm, D., & Lopes, M. P. (2019). Conveyed Leader PsyCap Predicting Leader Effectiveness Through Positive Energizing. Journal of Management, 45(4), 1689-1712. https://doi.org/10.1177/0149206317733510
b) Sajjad Hussain, Khurram Shahzad. (2022). Unpacking perceived organizational justice-organizational cynicism relationship: Moderating role of psychological capital. Asia Pacific Management Review, 27 (1), 10-17, 1029-3132. https://doi.org/10.1016/j.apmrv.2021.03.005”
9- Authors should present additional statistical analyses, such as convergent and discriminant analysis.
10- The theoretical implications of the study should be reinforced.
Author Response
We thank again the reviewer for the time spent in reviewing our work. Please, find below our answers to his/her comments.
Comment 1- The authors mention the following: “Our hypothesis is that both objective and perceived unfavorable workload characteristics are associated with worse anxiety and insomnia outcomes and that sex has a role in mediating this association. We also assume that the prevalence of anxiety and insomnia symptoms in PHRs is equal or higher when compared to the one reported in healthcare workers”. Nevertheless, the hypotheses proposed in the study should be clearly enumerated and adequately substantiated.
Response 1 - Thank you for highlighting this aspect. We have now clearly enumerated each hypothesis and provided the rationale behind them. (lines 138-154)
Comment 2 - Authors should make it clear whether there are mediating or moderating variables. On the other hand, the manuscript introduction should clarify the arguments for choosing mediation or moderation.
Response 2 - Thank you for pointing out this. Stating our hypotheses we mistakenly wrote “mediating” instead of “moderating”. We have now corrected this and specified the reason we are using a moderation analysis (131-135).
Comment 3 - The graphical representation of the relationships between variables is crucial. For better clarification, I leave the following example:
a) Maghsoud, F., Rezaei, M., Asgarian, F.S. et al. (2022). Workload and quality of nursing care: the mediating role of implicit rationing of nursing care, job satisfaction and emotional exhaustion by using structural equations modeling approach. BMC Nurs 21, 273. https://doi.org/10.1186/s12912-022-01055-1
Response 3 - Thank you for providing a practical example to support your suggestion. We have added a figure (Figure 1) illustrating the first two hypotheses at the end of the introduction and another figure (Figure 3) at the end of the results section to represent our findings. Given the number of variables considered (five for objective workload and two for perceived workload), along with the three different models we tested, each adjusted for different sets of covariates in both logistic and linear regression, we decided that a detailed depiction of each relationship would have resulted in an overly complex graph that might hinder readability. Instead, inspired by the example you provided, we chose to create a more general figure that effectively summarizes the main hypotheses and findings of our study.
Comment 4 - To facilitate comprehension, it is recommended that the descriptive statistics of the variables be presented in table format.
Response 4 - Thank you for your comment. Descriptive statistics of all the variables are now present in Table 1. Specifically, the table now includes also the summary statistics of the continuous scales used to measure anxiety and insomnia symptoms
Comment 5 - Authors should present the results of correlations between all variables under study.
Response 5 - Thank you for clarifying this point. We have now replaced the figure 2 with the results of the new collinearity analysis that accounts for all the variables under study and not only the independent variables
Comment 6 - Authors should present Cronbach’s alpha values ​​of the variables under study.
Response 6 - Thank you for this comment. Cronbach’s alpha values have been calculated and reported in the materials and methods section when presenting the questionnaires used for the purposes of our analyses.
Comment 7 - Authors should substantiate the procedures used to minimize common method bias based on the literature.
Response 7 - Thank you for your comment. We have described the procedures used to minimize common method bias (lines 504-516)
Comment 8 - I reiterate the following comment: “The authors should present a figure that reflects the results of the hypothesis test. On the other hand, authors must present graphs that reflect the influence of the moderating variable. For better clarification, I leave the following examples:
a) Rego, A., Yam, K. C., Owens, B. P., Story, J. S. P., Pina e Cunha, M., Bluhm, D., & Lopes, M. P. (2019). Conveyed Leader PsyCap Predicting Leader Effectiveness Through Positive Energizing. Journal of Management, 45(4), 1689-1712. https://doi.org/10.1177/0149206317733510
b) Sajjad Hussain, Khurram Shahzad. (2022). Unpacking perceived organizational justice-organizational cynicism relationship: Moderating role of psychological capital. Asia Pacific Management Review, 27 (1), 10-17, 1029-3132. https://doi.org/10.1016/j.apmrv.2021.03.005”
Response 8 - Thank you for your comment. As per comment number 3, we now included a figure representing the hypotheses (Figure 1) and a final figure (Figure 3) that represents the revised Figure 1 according to the results of our analyses. The graphs that reflect the influence of the moderating variable were already included in Supplementary materials (Figure S.1 and S.2)
Comment 9 - Authors should present additional statistical analyses, such as convergent and discriminant analysis.
Response 9 - Thank you for this valuable suggestion. As discussed in the limitations and strengths section, the questionnaires used in our study (GAD-2 for anxiety, ISI for insomnia, and items from the Work-Sense of Coherence and Work-Related Stress Questionnaire) have all been previously validated. Additionally, the GAD-2 and ISI are widely used in the literature and have consistently demonstrated validity in measuring their intended constructs across various populations, including healthcare workers. While we recognize that our study has ample room for development with additional analyses, performing a convergent and discriminant analysis falls outside the scope of our current objectives, which are specifically outlined in the hypotheses. We are aware that mental health in medical residents is a relatively underexplored research area, and we will certainly consider this suggestion in future work as we continue to expand on this topic.
Comment 10 - The theoretical implications of the study should be reinforced.
Response 10 - Thank you for the valuable suggestion. We have further strengthened the description of the theoretical implications of the study to emphasize the contribution to the knowledge on the relationship between workload and mental health, particularly in the context of residency training. Our results extend the information on occupational stress and mental health by showing how the characteristics of residency programmes and the academic and professional responsibilities of public health trainees can create important psychological pressures.
Reviewer 2 Report
Comments and Suggestions for Authors
I would like to thank the authors for considering the comments and changing the manuscript accordingly.
Author Response
We thank again the reviewer for his time spent in evaluating our work.